# THE CRAMÉR DISTANCE AS A SOLUTION TO BIASED WASSERSTEIN GRADIENTS

## ABSTRACT

The Wasserstein probability metric has received much attention from the machine learning community. Unlike the Kullback-Leibler divergence, which strictly measures change in probability, the Wasserstein metric reflects the underlying geometry between outcomes. The value of being sensitive to this geometry has been demonstrated, among others, in ordinal regression and generative modelling, and most recently in reinforcement learning. In this paper we describe three natural properties of probability divergences that we believe reflect requirements from machine learning: sum invariance, scale sensitivity, and unbiased sample gradients. The Wasserstein metric possesses the first two properties but, unlike the Kullback-Leibler divergence, does not possess the third. We provide empirical evidence suggesting this is a serious issue in practice. Leveraging insights from probabilistic forecasting we propose an alternative to the Wasserstein metric, the Cramér distance. We show that the Cramér distance possesses all three desired properties, combining the best of the Wasserstein and Kullback-Leibler divergences. We give empirical results on a number of domains comparing these three divergences. To illustrate the practical relevance of the Cramér distance we design a new algorithm, the Cramér Generative Adversarial Network (GAN), and show that it has a number of desirable properties over the related Wasserstein GAN.

## 1 INTRODUCTION

In machine learning, the Kullback-Leibler (KL) divergence is perhaps the most common way of assessing how well a probabilistic model explains observed data. Among the reasons for its popularity is that it is directly related to maximum likelihood estimation and is easily optimized. However, the KL divergence suffers from a significant limitation: it does not take into account how close two outcomes might be, but only their relative probability. This closeness can matter a great deal: in image modelling, for example, perceptual similarity is key (Rubner et al., 2000; Gao & Kleywegt, 2016). Put another way, the KL divergence cannot reward a model that "gets it almost right".

To address this limitation, researchers have turned to the Wasserstein metric, which does incorporate the underlying geometry between outcomes. The Wasserstein metric can be applied to distributions with non-overlapping supports, and has good out-of-sample performance (Esfahani & Kuhn, 2015). Yet, practical applications of the Wasserstein distance, especially in deep learning, remain tentative. In this paper we provide a clue as to why that might be: estimating the Wasserstein metric from samples yields biased gradients, and may actually lead to the wrong minimum. This precludes using stochastic gradient descent (SGD) and SGD-like methods, whose fundamental mode of operation is sample-based, when optimizing for this metric.

As a replacement we propose the Cramér distance (Székely, 2002; Rizzo & Székely, 2016), also known as the continuous ranked probability score in the probabilistic forecasting literature (Gneiting & Raftery, 2007). The Cramér distance, like the Wasserstein metric, respects the underlying geometry but also has unbiased sample gradients. To underscore our theoretical findings, we demonstrate a significant quantitative difference between the two metrics when employed in typical machine learning scenarios: categorical distribution estimation, regression, and finally image generation. In the latter case, we use a multivariate generalization of the Cramér distance, the energy distance (Székely, 2002), itself an instantiation of the MMD family of metrics (Gretton et al., 2012).

## 2 PROBABILITY DIVERGENCES AND METRICS

In this section we provide the notation to mathematically distinguish the Wasserstein metric (and later, the Cramér distance) from the Kullback-Leibler divergence and probability distances such as the total variation.

Let $P$ be a probability distribution over $\mathbb{R}$. When $P$ is continuous, we will assume it has density $\mu_P$. The expectation of a function $f : \mathbb{R} \to \mathbb{R}$ with respect to $P$ is

$$\underset{x \sim P}{\mathbb{E}} f(x) := \int_{-\infty}^{\infty} f(x) P(\mathrm{d}x) = \begin{cases} \int f(x) \mu_P(x) \mathrm{d}x & \text{if } P \text{ is continuous, and} \\ \sum f(x) P(x) & \text{if } P \text{ is discrete.} \end{cases}$$

We will suppose all expectations and integrals under consideration are finite. We will often associate $P$ to a random variable $X$, such that for a subset of the reals $A \subseteq \mathbb{R}$, we have $\Pr\{X \in A\} = P(A)$. The (cumulative) distribution function of $P$ is then

$$F_P(x) := \Pr\{X \le x\} = \int_{-\infty}^{x} P(dx).$$

Finally, the inverse distribution function of $P$, defined over the interval $(0, 1]$, is

$$F_P^{-1}(u) := \inf\{x : F_P(x) = u\}.$$

### 2.1 DIVERGENCES AND METRICS

Consider two probability distributions $P$ and $Q$ over $\mathbb{R}$. A *divergence* $\mathbf{d}$ is a mapping $(P, Q) \mapsto \mathbb{R}^+$ with $\mathbf{d}(P, Q) = 0$ if and only if $P = Q$ almost everywhere. A popular choice is the Kullback-Leibler (KL) divergence

$$\mathrm{KL}(P \,\|\, Q) := \int_{-\infty}^{\infty} \log \frac{P(\mathrm{d}x)}{Q(\mathrm{d}x)} P(\mathrm{d}x),$$

with $\mathrm{KL}(P \,\|\, Q) = \infty$ if $P$ is not absolutely continuous w.r.t. $Q$. The KL divergence, also called relative entropy, measures the amount of information needed to encode the change in probability from $Q$ to $P$ (Cover & Thomas, 1991).

A *probability metric* is a divergence which is also symmetric ($\mathbf{d}(P, Q) = \mathbf{d}(Q, P)$) and respects the triangle inequality: for any distribution $R$, $\mathbf{d}(P, Q) \le \mathbf{d}(P, R) + \mathbf{d}(R, Q)$. We will use the term *probability distance* to mean a symmetric divergence satisfying the relaxed triangle inequality $\mathbf{d}(P, Q) \le c\,[\mathbf{d}(P, R) + \mathbf{d}(R, Q)]$ for some $c \ge 1$.

We will first study the $p$-Wasserstein metrics $w_p$ (Dudley, 2002). For $1 \le p < \infty$, a practical definition is through the inverse distribution functions of $P$ and $Q$:

$$w_p(P, Q) := \left( \int_0^1 \left| F_P^{-1}(u) - F_Q^{-1}(u) \right|^p \mathrm{d}u \right)^{1/p}. \tag{1}$$

We will sometimes find it convenient to deal with the $p^{th}$ power of the metric, which we will denote by $w_p^p$; note that $w_p^p$ is not a metric proper, but is a probability distance.

We will be chiefly concerned with the 1-Wasserstein metric, which is most commonly used in practice. The 1-Wasserstein metric has a dual form which is theoretically convenient and which we mention here for completeness. Define $\mathbb{F}_\infty$ to be the class of 1-Lipschitz functions. Then

$$w_1(P, Q) := \sup_{f \in \mathbb{F}_\infty} \Big| \underset{x \sim P}{\mathbb{E}} f(x) - \underset{x \sim Q}{\mathbb{E}} f(x) \Big|. \tag{2}$$

This is a special case of the celebrated Monge-Kantorovich duality (Rachev et al., 2013), and is the integral probability metric (IPM) with function class $\mathbb{F}_\infty$ (Müller, 1997). We invite the curious reader to consult these two sources as a starting point on this rich topic.

### 2.2 PROPERTIES OF A DIVERGENCE

As noted in the introduction, the fundamental difference between the KL divergence and the Wasserstein metric is that the latter is sensitive not only to change in probability but also to the geometry of possible outcomes. To capture this notion we now introduce the concept of an *ideal divergence*.

Consider a divergence $\mathbf{d}$, and for two random variables $X, Y$ with distributions $P, Q$ write $\mathbf{d}(X, Y) := \mathbf{d}(P, Q)$. We say that $\mathbf{d}$ is *scale sensitive* (of order $\beta$), i.e. it has property **(S)**, if there exists a $\beta > 0$ such that for all $X, Y$, and a real value $c > 0$,

$$\mathbf{d}(cX, cY) \leq |c|^{\beta} \mathbf{d}(X, Y). \tag{S}$$

A divergence $\mathbf{d}$ has property **(I)**, i.e. it is *sum invariant*, if whenever $A$ is independent from $X, Y$

$$\mathbf{d}(A + X, A + Y) \leq \mathbf{d}(X, Y). \tag{I}$$

Following Zolotarev (1976), an ideal divergence $\mathbf{d}$ is one that possesses both (S) and (I).[1]

We can illustrate the sensitivity of ideal divergences to the value of outcomes by considering Dirac functions $\delta_x$ at different values of $x$. If $\mathbf{d}$ is scale sensitive of order $\beta = 1$ then the divergence $\mathbf{d}(\delta_0, \delta_{1/2})$ can be no more than half the divergence $\mathbf{d}(\delta_0, \delta_1)$. If $\mathbf{d}$ is *sum invariant*, then the divergence of $\delta_0$ to $\delta_1$ is equal to the divergence of the same distributions shifted by a constant $c$, i.e. of $\delta_c$ to $\delta_{1+c}$. As a concrete example of the importance of these properties, Bellemare et al. (2017) recently demonstrated the importance of ideal metrics in reinforcement learning, specifically their role in providing the contraction property of the distributional Bellman operator. In particular, the contraction modulus is $\gamma^{\beta}$, where $\gamma \in [0, 1)$ is a discount factor and $\beta$ is the scale sensitivity order.

In machine learning we often view the divergence $\mathbf{d}$ as a loss function. Specifically, let $Q_{\theta}$ be some distribution parametrized by $\theta$, and consider the loss $\theta \mapsto \mathbf{d}(P, Q_{\theta})$. We are interested in minimizing this loss, that is finding $\theta^* := \arg\min_{\theta} \mathbf{d}(P, Q_{\theta})$. We now describe a third property based on this loss, which we call *unbiased sample gradients*.

Let $\mathbf{X}_m := X_1, X_2, \dots, X_m$ be independent samples from $P$ and define the empirical distribution $\hat{P}_m := \hat{P}_m(\mathbf{X}_m) := \frac{1}{m} \sum_{i=1}^{m} \delta_{X_i}$ (note that $\hat{P}_m$ is a random quantity). From this, define the *sample loss* $\theta \mapsto \mathbf{d}(\hat{P}_m, Q_{\theta})$. We say that $\mathbf{d}$ has unbiased sample gradients when the expected gradient of the sample loss equals the gradient of the true loss for all $P$ and $m$:

$$\mathop{\mathbb{E}}_{\mathbf{X}_m \sim P} \nabla_{\theta} \mathbf{d}(\hat{P}_m, Q_{\theta}) = \nabla_{\theta} \mathbf{d}(P, Q_{\theta}). \tag{U}$$

The notion of unbiased sample gradients is ubiquitous in machine learning and in particular in deep learning. Specifically, if a divergence $\mathbf{d}$ does not possess (U) then minimizing it with stochastic gradient descent may not converge, or it may converge to the wrong minimum. Conversely, if $\mathbf{d}$ possesses (U) then we can guarantee that the distribution which minimizes the expected sample loss is $Q = P$. In the probabilistic forecasting literature, this makes $\mathbf{d}$ a *proper scoring rule* (Gneiting & Raftery, 2007).

We now characterize the KL divergence and the Wasserstein metric in terms of these properties. As it turns out, neither simultaneously possesses both (U) and (S).

**Proposition 1.** *The KL divergence has unbiased sample gradients (U), but is not scale sensitive (S).*

**Proposition 2.** *The Wasserstein metric is ideal (I, S), but does not have unbiased sample gradients.*

We will provide a proof of the bias in the sample Wasserstein gradients just below; the proof of the rest and later results are provided in the appendix.

## 3 BIAS IN THE SAMPLE GRADIENT ESTIMATES OF THE WASSERSTEIN DISTANCE

In this section we give theoretical evidence of serious issues with gradients of the sample Wasserstein loss. We will consider a simple Bernoulli distribution $P$ with parameter $\theta^* \in (0, 1)$, which we would like to estimate from samples. Our model is $Q_{\theta}$, a Bernoulli distribution with parameter $\theta$. We study the behaviour of stochastic gradient descent w.r.t. $\theta$ over the sample Wasserstein loss, specifically using the $p^{th}$ power of the metric (as is commonly done to avoid fractional exponents). Our results build on the example given by Bellemare et al. (2017), whose result is for $\theta^* = \frac{1}{2}$ and $m = 1$.

---

[1]Properties (S) and (I) are called *regularity* and *homogeneity* by Zolotarev; we believe our choice of terms is more machine learning-friendly.

Consider the estimate $\nabla_\theta w_p^p(\hat{P}_m, Q_\theta)$ of the gradient $\nabla_\theta w_p^p(P, Q_\theta)$. We now show that even in this simplest of settings, this estimate is biased, and we exhibit a lower bound on the bias for any value of $m$. Hence the Wasserstein metric does not have property (U). More worrisome still, we show that the minimum of the expected empirical Wasserstein loss $\theta \mapsto \mathbb{E}_{\mathbf{X}_m} \left[ w_p^p(\hat{P}_m, Q_\theta) \right]$ is not the minimum of the Wasserstein loss $\theta \mapsto w_p^p(P, Q_\theta)$. We then conclude that minimizing the sample Wasserstein loss by stochastic gradient descent may in general fail to converge to the minimum of the true loss.

**Theorem 1.** *Let $\hat{P}_m = \frac{1}{m} \sum_{i=1}^m \delta_{X_i}$ be the empirical distribution derived from $m$ independent samples $\mathbf{X}_m = X_1, \ldots, X_m$ drawn from a Bernoulli distribution $P$. Then for all $1 \leq p < \infty$,*

- **Non-vanishing minimax bias** *of the sample gradient. For any $m \geq 1$ there exists a pair of Bernoulli distributions $P, Q_\theta$ for which*

$$\left| \mathbb{E}_{\mathbf{X}_m \sim P} \left[ \nabla_\theta w_p^p(\hat{P}_m, Q_\theta) \right] - \nabla_\theta w_p^p(P, Q_\theta) \right| \geq 2e^{-2};$$

- **Wrong minimum** *of the sample Wasserstein loss. The minimum of the expected sample loss $\tilde{\theta} = \arg\min_\theta \mathbb{E}_{\mathbf{X}_m} \left[ w_p^p(\hat{P}_m, Q_\theta) \right]$ is in general different from the minimum of the true Wasserstein loss $\theta^* = \arg\min_\theta w_p^p(P, Q_\theta)$.*
- **Deterministic solutions** *to stochastic problems. For any $m \geq 1$, there exists a distribution $P$ with nonzero entropy whose sample loss is minimized by a distribution $Q_{\tilde{\theta}}$ with zero entropy.*

Taken as a whole, Theorem 1 states that we cannot in general minimize the Wasserstein loss using naive stochastic gradient descent methods. Although our result does not imply the lack of a stochastic optimization procedure for this loss,[2] we believe our result to be cause for concern. We leave as an open question whether an unbiased optimization procedure exists and is practical.

WASSERSTEIN BIAS IN THE LITERATURE

Our result is surprising given the prevalence of the Wasserstein metric in empirical studies. We hypothesize that this bias exists in published results and is an underlying cause of learning instability and poor convergence often remedied to by heuristic means. For example, Frogner et al. (2015) and Montavon et al. (2016) reported the need for a mixed KL-Wasserstein loss to obtain good empirical results, with the latter explicitly discussing the issue of wrong minima when using Wasserstein gradients.

We remark that our result also applies to the dual (2), since the losses are the same. This dual was recently considered by Arjovsky et al. (2017) as an alternative loss to the primal (1). The adversarial procedure proposed by the authors is a two time-scale process which first maximizes (2) w.r.t $f \in \mathbb{F}_\infty$ using $m$ samples, then takes a single stochastic gradient step w.r.t. $\theta$. Interestingly, this approach does seem to provide unbiased gradients as $m \to \infty$. However, the cost of a single gradient is now significantly higher, and for a fixed $m$ we conjecture that the minimax bias remains.

## 4 THE CRAMÉR DISTANCE

We are now ready to describe an alternative to the Wasserstein metric, the Cramér distance (Székely, 2002; Rizzo & Székely, 2016). As we shall see, the Cramér distance has the same appealing properties as the Wasserstein metric, but also provides us with unbiased sample gradients. As a result, we believe this underappreciated distance is an appealing alternative to the Wasserstein metric for many machine learning applications.

### 4.1 DEFINITION AND ANALYSIS

Recall that for two distributions $P$ and $Q$ over $\mathbb{R}$, their (cumulative) distribution functions are respectively $F_P$ and $F_Q$. The (squared) Cramér distance between $P$ and $Q$ is

$$l_2^2(P, Q) := \int_{-\infty}^{\infty} (F_P(x) - F_Q(x))^2 \mathrm{d}x.$$

---

[2]For example, if $P$ has finite support keeping track of the empirical distribution suffices.

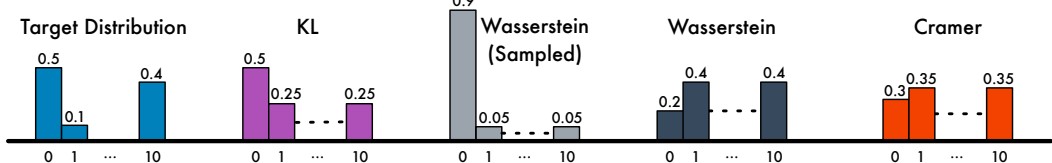

Figure 1: **Leftmost.** Target distribution. One outcome (10) is significantly more distant than the two others (0, 1). **Rest.** Distributions minimizing the divergences discussed in this paper, under the constraint $Q(1) = Q(10)$. Both Wasserstein metric and Cramér distance underemphasize $Q(0)$ to better match the cumulative distribution function. The sample Wasserstein loss result is for $m = 1$.

The Cramér distance is a Bregman divergence, and is a member of the $l_p$ family of divergences

$$l_p(P, Q) := \left( \int_{-\infty}^{\infty} |F_P(x) - F_Q(x)|^p \mathrm{d}x \right)^{1/p} .$$

The $l_p$ and Wasserstein metrics are identical at $p = 1$, but are otherwise distinct. As the following theorem shows, the Cramér distance possesses unique properties.

**Theorem 2.** *Consider two random variables $X, Y$, a random variable $A$ independent of $X, Y$, and a real value $c > 0$. Then for $1 \le p \le \infty$,*

**(I)** $l_p(A + X, A + Y) \le l_p(X, Y)$ **(S)** $l_p(cX, cY) \le |c|^{1/p} l_p(X, Y).$

*Furthermore, the Cramér distance has unbiased sample gradients. That is, given $\mathbf{X}_m := X_1, \ldots, X_m$ drawn from a distribution $P$, the empirical distribution $\hat{P}_m := \frac{1}{m} \sum_{i=1}^{m} \delta_{X_i}$, and a distribution $Q_\theta$,*

$$\mathop{\mathbb{E}}_{\mathbf{X}_m \sim P} \nabla_\theta l_2^2(\hat{P}_m, Q_\theta) = \nabla_\theta l_2^2(P, Q_\theta),$$

*and of all the $l_p$ distances, only the Cramér ($p = 2$) has this property.*

We conclude that the Cramér distance enjoys both the benefits of the Wasserstein metric and the SGD-friendliness of the KL divergence. Given the close similarity of the Wasserstein and $l_p$ metrics, it is truly remarkable that only the Cramér distance has unbiased sample gradients.

## 4.2 Comparison to the 1-Wasserstein Metric

To illustrate how the Cramér distance compares to the 1-Wasserstein metric, we consider modelling the discrete distribution $P$ depicted in Figure 1 (left). Since the trade-offs between metrics are only apparent when using an approximate model, we use an underparametrized discrete distribution $Q_\theta$ which assigns the same probability to $x = 1$ and $x = 10$. That is,

$$Q_\theta(0) := Q_\theta\{x = 0\} = \frac{1}{1 + 2e^\theta} \qquad Q_\theta(1) = Q_\theta(10) = \frac{e^\theta}{1 + 2e^\theta}.$$

Figure 1 depicts the distributions minimizing the various divergences under this parametrization. In particular, the Cramér solution is relatively close to the 1-Wasserstein solution. Furthermore, the minimizer of the sample Wasserstein loss ($m = 1$) clearly provides a bad solution (most of the mass is on 0). Note that, as implied by Theorem 1, the bias shown here would arise even if the distribution could be exactly represented.

To further show the impact of the Wasserstein bias we used gradient descent to minimize either the true or sample losses with a fixed step-size ($\alpha = 0.001$). In the stochastic setting, at each step we construct the empirical distribution $\hat{P}_m$ from $m$ samples (a Dirac when $m = 1$), and take a gradient step. We measure the performance of each method in terms of the true 1-Wasserstein loss.

Figure 2 (left) plots the resulting training curves in the 1-Wasserstein regime, with the KL and Cramér solutions indicated for reference. We first note that, compared to the KL solution, the Cramér solution has significantly smaller Wasserstein distance to the target distribution. Second, for

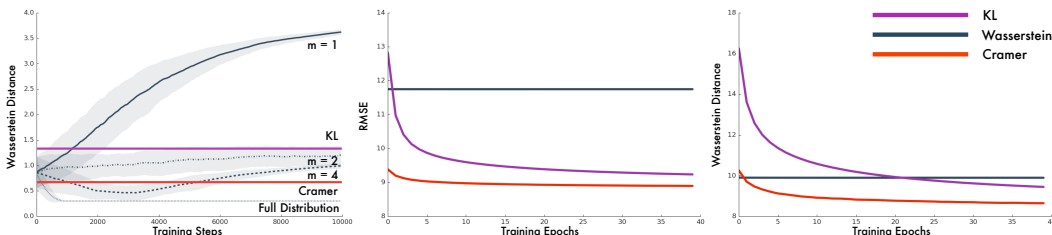

Figure 2: **Left.** Wasserstein distance in terms of SGD updates, minimizing the true or sample Wasserstein losses. Also shown are the distances for the KL and Cramér solutions. Results are averaged over 10 random initializations, with error-bands indicating one standard deviation. **Center.** Ordinal regression on the Year Prediction MSD dataset. Learning curves report RMSE on test set. **Right.** The same in terms of sample Wasserstein loss.

small sample sizes stochastic gradient descent fails to find reasonable solutions, and for $m = 1$ even converges to a solution worse than the KL minimizer. This small experiment highlights the cost incurred from minimizing the sample Wasserstein loss, and shows that increasing the sample size may not be sufficient to guarantee good behaviour.

### ORDINAL REGRESSION

We next trained a neural network in an ordinal regression task using either of the three divergences. The task we consider is the Year Prediction MSD dataset (Lichman, 2013). In this task, the model must predict the year a song was written (from 1922 to 2011) given a 90-dimensional feature representation. In our setting, this prediction takes the form of a probability distribution. We measure each method's performance on the test set (Figure 2) in two ways: root mean squared error (RMSE) – the metric minimized by Hernández-Lobato & Adams (2015) – and the sample Wasserstein loss. Full details on the experiment may be found in the appendix.

The results show that minimizing the sample Wasserstein loss results in significantly worse performance. By contrast, minimizing the Cramér distance yields the lowest RMSE and Wasserstein loss, confirming the practical importance of having unbiased sample gradients. Naturally, minimizing for one loss trades off performance with respect to the others, and minimizing the Cramér distance results in slightly higher negative log likelihood than when minimizing the KL divergence (Figure 7 in appendix). We conclude that, in the context of ordinal regression where outcome similarity plays an important role, the Cramér distance should be preferred over either KL or the Wasserstein metric.

## 5   MULTIVARIATE DISTRIBUTIONS

The *energy distance* (Székely, 2002) is a natural extension of the Cramér distance to the multivariate case. Let $P, Q$ be probability distributions over $\mathbb{R}^d$ and let $X, X'$ and $Y, Y'$ be independent random variables distributed according to $P$ and $Q$, respectively. The energy distance (sometimes called the squared energy distance, see e.g. Rizzo & Székely, 2016) is

$$\mathcal{E}(P, Q) := \mathcal{E}(X, Y) := 2 \, \mathbb{E} \, \|X - Y\|_2 - \mathbb{E} \, \|X - X'\|_2 - \mathbb{E} \, \|Y - Y'\|_2 . \tag{3}$$

Székely showed that, in the univariate case, $l_2^2(P, Q) = \frac{1}{2}\mathcal{E}(P, Q)$. Interestingly enough, the energy distance can also be written in terms of a difference of expectations. For

$$f^*(x) := \mathbb{E} \, \|x - Y'\|_2 - \mathbb{E} \, \|x - X'\|_2 ,$$

we find that

$$\mathcal{E}(X, Y) = \mathbb{E} \, f^*(X) - \mathbb{E} \, f^*(Y). \tag{4}$$

The energy distance is closely related to the distances known as maximum mean discrepancies (MMDs; Gretton et al., 2012); in particular, Sejdinovic et al. (2013) showed that the energy distance is equivalent to the squared MMD with kernel $k(x, y) = \|x\|_2 + \|y\|_2 - \|x - y\|_2$. Finally, we remark that $\mathcal{E}$ also possesses properties (I), (S), and (U) (proof in the appendix).

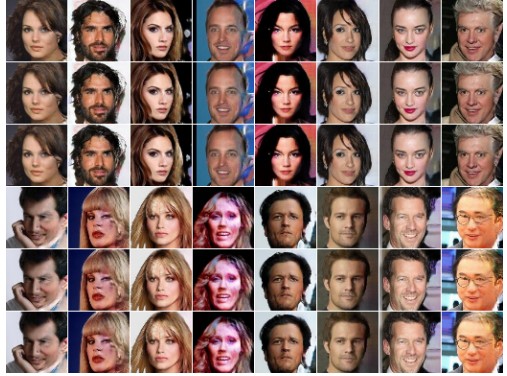 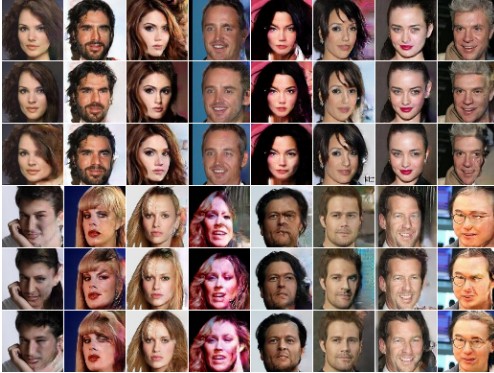

Figure 3: Generated right halves of the faces for WGAN-GP (left) and Cramér GAN (right). The given left halves are from CelebA 64x64 validation set (Liu et al., 2015).

---

**Algorithm 1:** Cramér GAN Losses.

**Parameter.** Gradient penalty coefficient $\lambda$.
Sample $x_r \sim P$, $x_g, x'_g \sim Q$, $\epsilon \sim \text{Uniform}(0, 1)$.
Interpolate real and generated samples:
$\hat{x} = \epsilon x_r + (1 - \epsilon) x_g$
Sample generator loss (12):
$\hat{L}_g = \|h(x_r) - h(x_g)\|_2 + \|h(x_r) - h(x'_g)\|_2$
$\qquad - \|h(x_g) - h(x'_g)\|_2$
Sample surrogate generator loss (13) and critic loss:
$\tilde{L}_s(u, v) = \|h(x_r) - h(v)\|_2 - \|h(x_r)\|_2$
$\qquad\qquad - \|h(u) - h(v)\|_2 + \|h(u)\|_2$
$L_s = \frac{1}{2}\left[ \tilde{L}_s(x_g, x'_g) + \tilde{L}_s(x'_g, x_g) \right]$
$f(x) = \|h(x) - h(x_g)\|_2 - \|h(x) - h(x_r)\|_2$
$L_{critic} = -L_s + \lambda(\|\nabla_{\hat{x}} f(\hat{x})\|_2 - 1)^2$

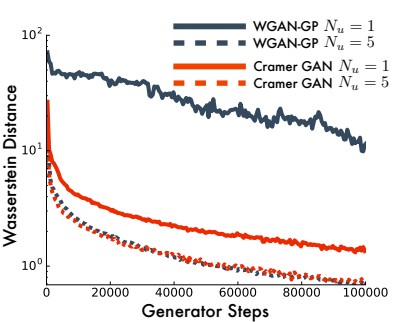

Figure 4: Approximate Wasserstein distances between CelebA test set and the generators. $N_u$ is the number critic updates per generator update.

## 5.1 CRAMÉR GAN

We now consider the Generative Adversarial Networks (GAN) framework (Goodfellow et al., 2014), in particular issues arising in the Wasserstein GAN (Arjovsky et al., 2017), and propose a better GAN based on the Cramér distance. A GAN is composed of a generative model $Q$ (in our experiments, over images), called the *generator*, a target source $P$, and a trainable loss function called a discriminator or *critic*. GANs are particularly interesting because we can establish a direct comparison between the two distances. Our choice of name reflects this fact, and we prefer *Cramér GAN* to the perhaps more technically correct, but less palatable *Energy Distance GAN*. In theory, the Wasserstein GAN algorithm requires training the critic until convergence, but this is rarely achievable: we would require a critic that is a very powerful network to approximate the Wasserstein distance well (Arora et al., 2017). Simultaneously, training this critic to convergence would overfit the empirical distribution of the training set, which is undesirable.

Our proposed loss function allows for useful learning with imperfect critics by combining the energy distance with a transformation function $h : \mathbb{R}^d \to \mathbb{R}^k$, where $d$ is the input dimensionality and $k = 256$ in our experiments. The generator then seeks to minimize the energy distance of the transformed variables $\mathcal{E}(h(X), h(Y))$, where $X$ is a real sample and $Y$ is a generated sample. The critic itself seeks to maximize this same distance by changing the parameters of $h$, subject to a soft constraint (the gradient penalty used by Gulrajani et al., 2017). Specifically, the critic maximizes a surrogate loss whose gradient can be estimated from a single real sample. The Cramér GAN losses are summarized in Algorithm 1, with additional design choices detailed in Appendix C.

We note that MMDs such as the energy distance have in the last year become an appealing tool for training GANs. Among others, the squared MMD is used within Generative Moment Matching Networks (Li et al., 2015; Dziugaite et al., 2015); Bouchacourt et al. (2016) trained a model to minimize the energy distance for hand pose estimation. Our use of the tranformation $h(x)$ reflects our anecdotal finding that the direct minimization of the energy distance over raw images does not work well (see Figure 10 in appendix). Similar findings can be found in the work of Mroueh et al. (2017) and the independently developed MMD GAN (Li et al., 2017), which additionally uses an auto-encoder loss to make the transformation injective.

The Cramér GAN we present here complements our comparison of the Wasserstein and Cramér distance from previous sections. At the same time, our experiments also provide novel GAN-related contributions, including the ability to perform conditional modelling using a surrogate generator loss, which lets us train the critic even when only one independent sample from $P$ is available. We note also that in our experiments, $\|x - y\|_2$ distances were more stable than distances generated by Gaussian or Laplacian kernels.

## 5.2 CRAMÉR GAN EXPERIMENTS

We now show that, compared to the improved Wasserstein GAN (WGAN-GP) of Gulrajani et al. (2017), the Cramér GAN leads to more stable learning and increased diversity in the generated samples. In both cases we train generative models that predict the right half of an image given the left half; samples from unconditional models are provided in the appendix (Figure 10). The dataset we use here is the CelebA $64 \times 64$ dataset (Liu et al., 2015) of celebrity faces.

**Increased diversity.** In our first experiment, we compare the qualitative diversity of completed faces by showing three sample completions generated by either model given the left half of a validation set image (Figure 3). We observe that the completions produced by WGAN-GP are almost deterministic. Our findings echo those of Isola et al. (2016), who observed that *"the generator simply learned to ignore the noise."* By contrast, the completions produced by Cramér GAN are fairly diverse, including different hairstyles, accessories, and backgrounds. We view this lack of diversity in WGAN-GP as undesirable given that the main requirement of a generative model is that it should provide a variety of outputs.

Theorem 1 provides a clue as to what may be happening here. We know that minimizing the sample Wasserstein loss will find the wrong minimum. In particular, when the target distribution has low entropy, the sample Wasserstein minimizer may actually be a deterministic distribution. But a good generative model of images *must* lie in this "almost deterministic" regime, since the space of natural images makes up but a fraction of all possible pixel combinations and hence there is little per-pixel entropy. We hypothesize that the increased diversity in the Cramér GAN comes exactly from learning these almost deterministic predictions.

**More stable learning.** In a second experiment, we varied the number of critic updates ($N_u$) per generator update. To compare performance between the two architectures, we measured the loss computed by an independent WGAN-GP critic trained on the validation set, following a similar evaluation previously done by Danihelka et al. (2017). Figure 4 shows the independent Wasserstein critic distance between each generator and the test set during the course of training. Echoing our results with the toy experiment and ordinal regression, the plot shows that when a single critic update is used, WGAN-GP performs particularly poorly. We note that additional critic updates also improve Cramér GAN. This indicates that it is helpful to keep adapting the $h(x)$ transformation.

## 6 CONCLUSION

There are many situations in which the KL divergence, which is commonly used as a loss function in machine learning, is not suitable. The desirable alternatives, as we have explored, are the divergences that are ideal and allow for unbiased estimators: they allow geometric information to be incorporated into the optimization problem; because they are scale-sensitive and sum-invariant, they possess the convergence properties we require for efficient learning; and the correctness of their sample gradients means we can deploy them in large-scale optimization problems. Among open questions, we mention deriving an unbiased estimator that minimizes the Wasserstein distance, and variance analysis and reduction of the Cramér distance gradient estimate.

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

# A  PROOFS

## A.1  PROPERTIES OF A DIVERGENCE

*Proof (Proposition 1 and 2).* The statement regarding (U) for the KL divergence is well-known, and forms the basis of most stochastic gradient algorithms for classification. Chung & Sobel (1987) have shown that the total variation does not have property (S); by Pinsker's inequality, it follows that the same holds for the KL divergence. A proof of (I) and (S) for the Wasserstein metric is given by Bickel & Freedman (1981), while the lack of (U) is shown in the proof of Theorem 1. □

## A.2  BIASED ESTIMATE

*Proof (Theorem 1).* **Minimax bias:** Consider $P = \mathcal{B}(\theta^*)$, a Bernoulli distribution of parameter $\theta^*$ and $Q_\theta = \mathcal{B}(\theta)$ a Bernoulli of parameter $\theta$. The empirical distribution $\hat{P}_m$ is a Bernoulli with parameter $\hat{\theta} := \frac{1}{m} \sum_{i=1}^m X_i$. Note that with $P$ and $Q_\theta$ both Bernoulli distributions, the $p^{th}$ powers of the $p$-Wasserstein metrics are equal, i.e. $w_1(P, Q_\theta) = w_p^p(P, Q_\theta)$. This gives us an easy way to prove the stronger result that all $p$-Wasserstein metrics have biased sample gradients. The gradient of the loss $w_p^p(P, Q_\theta)$ is, for $\theta \neq \theta^*$,

$$g := \nabla w_p^p(P, Q_\theta) = \nabla \left[ |\theta^* - \theta| \right] = \text{sgn}(\theta - \theta^*),$$

and similarly, the gradient of the sample loss is, for $\theta \neq \hat{\theta}$,

$$\hat{g} := \nabla w_p^p(\hat{P}_m, Q_\theta) = \nabla \left[ |\hat{\theta} - \theta| \right] = \text{sgn}(\theta - \hat{\theta}).$$

Notice that this estimate is biased for any $m \geq 1$ since

$$\mathbb{E}\,\hat{g} = 2 \Pr\{\hat{\theta} < \theta\} - 1, \tag{5}$$

which is different from $g$ for any $\theta^* \in (0, 1)$. In particular for $m = 1$, $\mathbb{E}_P\,\hat{g} = 1 - 2\theta^*$ does not depend on $\theta$, thus a gradient descent using a one-sample gradient estimate has no chance of minimizing the Wasserstein loss as it will converge to either 1 or 0 instead of $\theta^*$.

Now observe that for $m \geq 2$, and any $\theta > \frac{m-1}{m}$,

$$\Pr\{\hat{\theta} < \theta\} = \Pr\{\exists i \text{ s.t. } X_i = 0\} = 1 - (\theta^*)^m,$$

and therefore

$$\mathbb{E}\,\hat{g} = 1 - 2(\theta^*)^m.$$

Taking $\theta^* = \frac{m-1}{m}$, we find that

$$g - \mathbb{E}\,\hat{g} = 1 - [1 - 2(\theta^*)^m] = 2\left(1 - \frac{1}{m}\right)^m \geq 2e^{-2}.$$

Thus for any $m$, there exists $P = \mathcal{B}(\theta^*)$ and $Q_\theta = \mathcal{B}(\theta)$ with $\theta^* = \frac{m-1}{m} < \theta < 1$ such that the bias $g - \mathbb{E}\,\hat{g}$ is lower-bounded by a numerical constant. Thus the *minimax bias* does not vanish with the number of samples $m$.

Notice that a similar argument holds for $\theta^*$ and $\theta$ being close to 0. In both situations where $\theta^*$ is close to 0 or 1, the bias is non vanishing when $|\theta^* - \theta|$ is of order $\frac{1}{m}$. However this is even worse when $\theta^*$ is away from the boundaries. For example chosing $\theta^* = \frac{1}{2}$, we can prove that the bias is non vanishing even when $|\theta^* - \theta|$ is (only) of order $\frac{1}{\sqrt{m}}$.

Indeed, using the anti-concentration result of Veraar (2010) (Proposition 2), we have that for a sequence $Y_1, \ldots, Y_m$ of Rademacher random variables (i.e. $+/-1$ with equal probability),

$$\Pr\left(\frac{1}{n} \sum_{i=1}^m Y_i \geq \epsilon\right) \geq (1 - m\epsilon^2)^2/3.$$

This means that for samples $X_1, \ldots, X_m$ drawn from a Bernoulli $\mathcal{B}(\theta^* = \frac{1}{2})$ (i.e., $Y_i = 2X_i - 1$ are Rademacher), we have

$$\Pr\left(\hat{\theta} \geq \theta^* + \epsilon/2\right) \geq (1 - m\epsilon^2)^2/3,$$

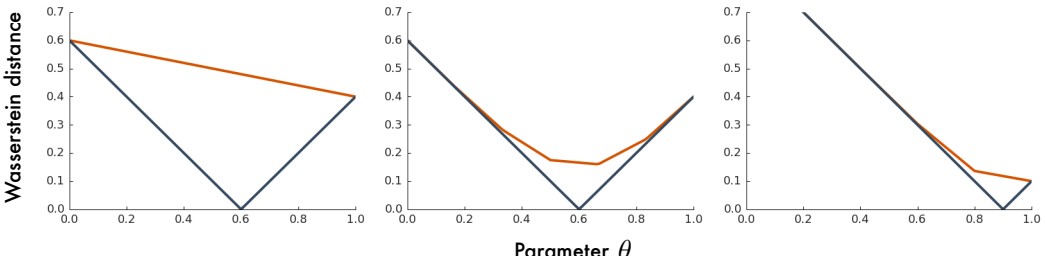

Figure 5: Wasserstein loss (black curve) $\theta \mapsto |\theta^* - \theta|$ versus expected sample Wasserstein loss (red curve) $\theta \mapsto \mathbb{E}[|\hat{\theta} - \theta|]$, for different values of $m$ and $\theta^*$ and $p = 1$. **Left:** $m = 1$, $\theta^* = 0.6$. A stochastic gradient using a one-sample Wasserstein gradient estimate will converge to $1$ instead of $\theta^*$. **Middle:** $m = 6$, $\theta^* = 0.6$. The minimum of the expected sample Wasserstein loss is the median of $\hat{\theta}$ which is here $\tilde{\theta} = \frac{2}{3} \neq \theta^* = 0.6$. **Right:** $m = 5$, $p = 0.9$. The minimum of the expected sample Wasserstein is $\tilde{\theta} = 1$ and not $\theta^* = 0.9$.

thus for $1/2 = \theta^* < \theta < \theta^* + 1/\sqrt{8m}$ we have the following lower bound on the bias:

$$g - \mathbb{E}\,\hat{g} = 2\Pr\left(\hat{\theta} \geq \theta\right) \geq 1/6.$$

Thus the bias is lower-bounded by a constant (independent of $m$) when $\theta^* = \frac{1}{2}$ and $|\theta^* - \theta| = O(1/\sqrt{m})$.

**Wrong minimum:** From (5), we deduce that a stochastic gradient descent algorithm based on the sample Wasserstein gradient will converge to a $\tilde{\theta}$ such that $\Pr\{\hat{\theta} < \tilde{\theta}\} = \frac{1}{2}$, i.e., $\tilde{\theta}$ is the median of the distribution over $\hat{\theta}$, whereas $\theta^*$ is the mean of that distribution. Since $\hat{\theta}$ follows a (normalized) binomial distribution with parameters $m$ and $\theta^*$, we know that the median $\tilde{\theta}$ and the mean $\theta^*$ do not necessarily coincide, and can actually be as far as $\frac{1}{2m}$-away from each other. For example for any odd $m$ and any $\theta^* \in \left(\frac{1}{2}, \frac{1}{2} - \frac{1}{2m}\right)$ the median is $\theta^* - \frac{1}{2m}$.

It follows that the minimum of the expected sample Wasserstein loss (the fixed point of the stochastic gradient descent using the sample Wasserstein gradient) is different from the minimum of the true Wasserstein loss:

$$\arg\min_{\theta} \mathbb{E}[w_p^p(\hat{P}_m, Q_\theta)] \neq \arg\min_{\theta}[w_p^p(P, Q_\theta)].$$

This is illustrated in Figure 5.

Notice that the fact that the minima of these losses differ is worrisome as it means that minimizing the sample Wasserstein loss using (finite) samples will not converge to the correct solution.

**Deterministic solutions:** Consider the specific case where $(1/2)^{1/n} < \theta^* < 1$ (illustrated in the right plot of Figure 5). Then the expected sample gradient $\nabla \mathbb{E}[w_p^p(\hat{P}_m, Q_{\theta^*})] = \mathbb{E}\hat{g} = 1 - 2(\theta^*)^n < 0$ for any $\theta$, so a gradient descent algorithm will converge to $1$ instead of $\theta^*$. Notice that a symmetric argument applies for $\theta^*$ close to $0$.

In this simple example, minimizing the sample Wasserstein loss may lead to degenerate solutions (i.e., deterministic) when our target distributions have low (but not zero) entropy. $\qquad\square$

### A.3 CONSISTENCY OF THE SAMPLE 1-WASSERSTEIN GRADIENT

We provide an additional result here showing that the sample 1-Wasserstein gradient converges to the true gradient as $m \to \infty$.

**Theorem 3.** *Let $P$ and $Q_\theta$ be probability distributions, with $Q_\theta$ parametrized by $\theta$. Assume that the set $\{x \in X,$ such that $F_P(x) = F_{Q_\theta}(x)\}$ has measure zero, and that for any $x \in X$, the map $\tilde{\theta} \mapsto F_{Q_{\tilde{\theta}}}(x)$ is differentiable in a neighborhood $\mathcal{V}(\theta)$ of $\theta$ with a uniformly bounded derivative*

*(for $\tilde{\theta} \in \mathcal{V}(\theta)$ and $x \in X$). Let $\hat{P}_m = \frac{1}{m}\sum_{i=1}^{m}\delta_{X_i}$ be the empirical distribution derived from $m$ independent samples $X_1, \ldots, X_m$ drawn from $P$. Then*

$$\lim_{m\to\infty} \nabla w_1(\hat{P}_m, Q_\theta) = \nabla w_1(P, Q_\theta), \text{ almost surely.}$$

We note that the measure requirement is strictly to keep the proof simple, and does not subtract from the generality of the result.

*Proof.* Let $\nabla := \nabla_\theta$. Since $p = 1$ the Wasserstein distance $w_1(P, Q)$ measures the area between the curves defined by the distribution function of $P$ and $Q$, thus $w_1(P, Q) = l_1(P, Q) = \int |F_P(x) - F_Q(x)| dx$ and

$$
\begin{aligned}
\nabla w_1(P, Q_\theta) &= \lim_{\Delta\to0} \frac{w_1(P, Q_{\theta+\Delta}) - w_1(P, Q_\theta)}{\Delta} \\
&= \lim_{\Delta\to0} \int \frac{1}{\Delta}\Big( |F_P(x) - F_{Q_{\theta+\Delta}}(x)| - |F_P(x) - F_{Q_\theta}(x)| \Big) dx.
\end{aligned}
$$

Now since we have assumed that for any $x \in X$, the map $\theta \mapsto F_{Q_\theta}(x)$ is differentiable in a neighborhood $\mathcal{V}(\theta)$ of $\theta$ and its derivative is uniformly (over $\mathcal{V}(\theta)$ and $x$) bounded by $M$, we have

$$\frac{1}{\Delta}\Big| |F_P(x) - F_{Q_{\theta+\Delta}}(x)| - |F_P(x) - F_{Q_\theta}(x)| \Big| \leq \frac{1}{\Delta}|F_{Q_{\theta+\Delta}}(x) - F_{Q_\theta}(x)| \leq M.$$

Thus the dominated convergence theorem applies and

$$
\begin{aligned}
\nabla w_1(P, Q_\theta) &= \int \lim_{\Delta\to0} \frac{1}{\Delta}\Big( |F_P(x) - F_{Q_{\theta+\Delta}}(x)| - |F_P(x) - F_{Q_\theta}(x)| \Big) dx \\
&= \int \nabla|F_P(x) - F_{Q_\theta}(x)| dx \\
&= \int \text{sgn}\big(F_P(x) - F_{Q_\theta}(x)\big) \nabla F_{Q_\theta}(x) dx,
\end{aligned}
$$

since we have assumed that the set of $x \in X$ such that $F_P(x) = F_{Q_\theta}(x)$ has measure zero.

Now, using the same argument for $w_1(\hat{P}_m, Q_\theta)$ we deduce that

$$\nabla w_1(\hat{P}_m, Q_\theta) = \int \underbrace{\lim_{\Delta\to0} \frac{1}{\Delta}\Big( |F_{\hat{P}_m}(x) - F_{Q_{\theta+\Delta}}(x)| - |F_{\hat{P}_m}(x) - F_{Q_\theta}(x)| \Big)}_{A(x)} dx.$$

Let us decompose this integral over $X$ as the sum of two integrals, one over $X \setminus \Omega_m$ and the other one over $\Omega_m$, where $\Omega_m = \{x \in X, F_{\hat{P}_m}(x) = F_{Q_\theta}(x)\}$. We have

$$\int_{X\setminus\Omega_m} A(x)dx = \int_{X\setminus\Omega_m} \text{sgn}\big(F_{\hat{P}_m}(x) - F_{Q_\theta}(x)\big)\nabla F_{Q_\theta}(x)dx,$$

and

$$
\begin{aligned}
\Big| \int_{\Omega_m} A(x)dx \Big| &\leq \int_{\Omega_m} \lim_{\Delta\to0}\frac{1}{\Delta}\Big( |F_{Q_{\theta+\Delta}}(x) - F_{Q_\theta}(x)| \Big)dx \\
&\leq M|\Omega_m|.
\end{aligned}
$$

Now from the strong law of large numbers, we have that for any $x$, the empirical cumulative distribution function $F_{\hat{P}_m}(x)$ converges to the cumulative distribution $F_P(x)$ almost surely. We deduce that $\Omega_m$ converges to the set $\{x, F_P(x) = F_{Q_\theta}(x)\}$ which has measure zero, thus $|\Omega_m| \to 0$ and

$$\lim_{m\to\infty} \nabla w_1(\hat{P}_m, Q_\theta) = \lim_{m\to\infty} \int_X \text{sgn}\big(F_{\hat{P}_m}(x) - F_{Q_\theta}(x)\big)\nabla F_{Q_\theta}(x)dx.$$

Now, since $|\nabla F_{Q_\theta}(x)| \le M$, we can use once more the dominated convergence theorem to deduce that

$$
\begin{aligned}
\lim_{m \to \infty} \nabla w_1(\hat{P}_m, Q_\theta) &= \int_X \lim_{m \to \infty} \mathrm{sgn}\big(F_{\hat{P}_m}(x) - F_{Q_\theta}(x)\big) \nabla F_{Q_\theta}(x) dx \\
&= \int_X \mathrm{sgn}\big(F_P(x) - F_{Q_\theta}(x)\big) \nabla F_{Q_\theta}(x) dx \\
&= \nabla w_1(P, Q_\theta). \qquad \qquad \square
\end{aligned}
$$

The following lemma will be useful in proving that the Cramér distance has property (U).

**Lemma 1.** *Let* $\mathbf{X}_m := X_1, \dots, X_m$ *be independent samples from* $P$, *and let* $\hat{P}_m := \frac{1}{m} \sum_i \delta_{X_i}$. *Then*

$$
\mathop{\mathbb{E}}_{\mathbf{X}_m \sim P} F_{\hat{P}_m}(x) = F_P(x).
$$

*Proof.* Because the $X_i$'s are independent,

$$
F_{\hat{P}_m}(x) = \int_{-\infty}^x \hat{P}_m(\mathrm{d}x) = \frac{1}{m} \sum_{i=1}^m \mathbb{I}\left[X_i \le x\right].
$$

Now, taking the expectation w.r.t. $\mathbf{X}_m$,

$$
\begin{aligned}
\mathop{\mathbb{E}}_{\mathbf{X}_m \sim P} F_{\hat{P}_m}(x) &= \mathop{\mathbb{E}}_{\mathbf{X}_m \sim P} \frac{1}{m} \sum_{i=1}^m \mathbb{I}\left[X_i \le x\right] \\
&= \frac{1}{m} \sum_{i=1}^m \mathop{\mathbb{E}}_{X_i \sim P} \mathbb{I}\left[X_i \le x\right] \\
&= \frac{1}{m} \sum_{i=1}^m \Pr\{X_i \le x\} \\
&= F_P(x),
\end{aligned}
$$

since the $X_i$ are identically distributed according to $P$. $\qquad \square$

*Proof (Theorem 2).* Like the Wasserstein metrics, the $l_p$ metrics have dual forms as integral probability metrics (see Dedecker & Merlevède, 2007, for a proof):

$$
l_p(P, Q) = \sup_{f \in \mathbb{F}_q} \left| \mathop{\mathbb{E}}_{x \sim P} f(x) - \mathop{\mathbb{E}}_{x \sim Q} f(x) \right|, \tag{6}
$$

where $\mathbb{F}_q := \{f : f \text{ is absolutely continuous}, \left\| \frac{\mathrm{d}f}{\mathrm{d}x} \right\|_q \le 1\}$ and $q$ is the conjugate exponent of $p$, i.e. $p^{-1} + q^{-1} = 1$.[3] We will use this dual form below.

We will prove that $l_p$ has properties (I) and (S) for $p \in [1, \infty)$; the case $p = \infty$ follows by a similar argument. Begin by observing that

$$
\begin{aligned}
F_{cX}(x) &= Pr\{cX \le x\} \\
&= Pr\left\{X \le \frac{x}{c}\right\} \\
&= F_X\left(\frac{x}{c}\right).
\end{aligned}
$$

Then we may rewrite $l_p^p(cX, cY)$ as

$$
\begin{aligned}
l_p^p(cX, cY) &= \int_{-\infty}^\infty \left| F_X\left(\frac{x}{c}\right) - F_Y\left(\frac{x}{c}\right) \right|^p \mathrm{d}x \\
&\overset{(a)}{=} c \int_{-\infty}^\infty \left| F_X(z) - F_Y(z) \right|^p \mathrm{d}z,
\end{aligned}
$$

---

[3] This relationship is the reason for the notation $\mathbb{F}_\infty$ in the definition the dual of the 1-Wasserstein (2).

where $(a)$ uses a change of variables $z = x/c$. Taking both sides to the power $1/p$ proves that the $l_p$ metric possesses property (S) of order $1/p$. For (I), we use the IPM formulation (6):

$$l_p(A + X, A + Y) = \sup_{f \in \mathcal{F}_q} \left| \underset{A+X}{\mathbb{E}} f(x) - \underset{A+Y}{\mathbb{E}} f(y) \right|$$

$$\overset{(a)}{=} \sup_{f \in \mathcal{F}_q} \left| \mathbb{E}_A \mathbb{E}_X f(x + a) - \mathbb{E}_A \mathbb{E}_Y f(y + a) \right|$$

$$= \sup_{f \in \mathcal{F}_q} \left| \mathbb{E}_A \left[ \mathbb{E}_X f(x + a) - \mathbb{E}_Y f(y + a) \right] \right|$$

$$\overset{(b)}{\leq} \mathbb{E}_A \sup_{f \in \mathcal{F}_q} \left| \mathbb{E}_X f(x + a) - \mathbb{E}_Y f(y + a) \right|,$$

where $(a)$ is by independence of $A$ and $X$, $Y$, and $(b)$ is by Jensen's inequality. Next, recall that $\mathcal{F}_q$ is the set of absolutely continuous functions whose derivative has bounded $L_q$ norm. Hence if $f \in \mathcal{F}_q$, then also for all $a$ the translate $g_a(x) := f(x + a)$ is also in $\mathcal{F}_q$. Therefore,

$$l_p(A + X, A + Y) \leq \mathbb{E}_A \sup_{f \in \mathcal{F}_q} \left| \mathbb{E}_X f(x + a) - \mathbb{E}_Y f(y + a) \right|$$

$$= \mathbb{E}_A \sup_{g \in \mathcal{F}_q} \left| \mathbb{E}_X g(x) - \mathbb{E}_Y g(y) \right|$$

$$= \sup_{g \in \mathcal{F}_q} \left| \mathbb{E}_X g(x) - \mathbb{E}_Y g(y) \right|$$

$$= l_p(X, Y). \qquad \square$$

Now, to prove **(U)**. Here we make use of the introductory requirement that "all expectations under consideration are finite." Specifically, we require that the mean under $P$, $\mathbb{E}_{x \sim P}[x]$, is well-defined and finite, and similarly for $Q_\theta$. In this case,

$$\underset{x \sim P}{\mathbb{E}}[x] = \int_0^\infty (1 - F_P(x)) \mathrm{d}x - \int_{-\infty}^0 F_P(x) \mathrm{d}x. \tag{7}$$

This mild requirement guarantees that the tails of the distribution function $F_P$ are light enough to avoid infinite Cramér distances and expected gradients (a similar condition was set by Dedecker & Merlevède (2007)). Now, by definition,

$$\nabla_\theta l_2^2(P, Q_\theta) = \nabla_\theta \int_{-\infty}^\infty \left( F_{Q_\theta}(x) - F_P(x) \right)^2 \mathrm{d}x$$

$$\overset{(a)}{=} \int_{-\infty}^\infty \nabla_\theta \left( F_{Q_\theta}(x) - F_P(x) \right)^2 \mathrm{d}x$$

$$= \int_{-\infty}^\infty 2 \left( F_{Q_\theta}(x) - F_P(x) \right) \nabla_\theta F_{Q_\theta}(x) \mathrm{d}x$$

$$\overset{(b)}{=} \int_{-\infty}^\infty 2 \left( F_{Q_\theta}(x) - \mathbb{E}_{\mathbf{X}_m} F_{\hat{P}_m}(x) \right) \nabla_\theta F_{Q_\theta}(x) \mathrm{d}x$$

$$= \int_{-\infty}^\infty 2 \mathbb{E}_{\mathbf{X}_m} \left( F_{Q_\theta}(x) - F_{\hat{P}_m}(x) \right) \nabla_\theta F_{Q_\theta}(x) \mathrm{d}x$$

$$\overset{(c)}{=} \mathbb{E}_{\mathbf{X}_m} \int_{-\infty}^\infty 2 \left( F_{Q_\theta}(x) - F_{\hat{P}_m}(x) \right) \nabla_\theta F_{Q_\theta}(x) \mathrm{d}x$$

$$= \mathbb{E}_{\mathbf{X}_m} \nabla_\theta l_2^2(\hat{P}_m, Q_\theta),$$

where (a) follows from the hypothesis (7) (the convergence of the squares follows from the convergence of the ordinary values), (b) follows from Lemma 1 and (c) follows from Fubini's theorem, again invoking (7).

Finally, we prove that **of all the $l_p^p$ distances ($1 \leq p \leq \infty$) only the Cramér distance, $l_2^2$, has the (U) property.**

Without loss of generality, let us suppose $P$ is not a Dirac, and further suppose that for any $\mathbf{X}_m \sim P$, $F_{Q_\theta}(x) \geq F_{\hat{P}_m}(x)$ everywhere. For example, when $Q_\theta$ has bounded support we can take $P$ to be a sufficiently translated version of $Q_\theta$, such that the two distributions' supports do not overlap.

We have already established that the 1-Wasserstein does not have the (U) property, and is equivalent to $l_p^p$ for $p = 1$. We will thus assume that $p > 1$, and also that $p < \infty$, the latter being recovered through standard limit arguments. Begin with the gradient for $l_p^p(P, Q_\theta)$,

$$
\begin{aligned}
\nabla_\theta l_p^p(P, Q_\theta) &= \nabla_\theta \int_{-\infty}^\infty \left| F_{Q_\theta}(x) - F_P(x) \right|^p \mathrm{d}x \\
&\overset{(a)}{=} p \int_{-\infty}^\infty \left( F_{Q_\theta}(x) - F_P(x) \right)^{p-1} \nabla_\theta F_{Q_\theta}(x) \mathrm{d}x \\
&= p \int_{-\infty}^\infty \phi_p(F_{Q_\theta}(x) - F_P(x)) \nabla_\theta F_{Q_\theta}(x) \mathrm{d}x \\
&= p \int_{-\infty}^\infty \phi_p \left( \mathbb{E}_{\mathbf{X}_m} \left( F_{Q_\theta}(x) - F_{\hat{P}_m}(x) \right) \right) \nabla_\theta F_{Q_\theta}(x) \mathrm{d}x,
\end{aligned}
$$

for $\phi_p(z) = z^{p-1}$; in (a) we used the same argument as in Theorem 3.

Now, $\phi_p$ is convex on $[0, \infty)$ when $p \geq 2$ and concave on the same interval when $1 < p < 2$. From Jensen's inequality we know that for a convex (concave) function $\phi$ and a random variable $Z$, $\mathbb{E}\,\phi(Z)$ is greater than (less than) or equal to $\phi(\mathbb{E}\,Z)$, with equality if and only if $\phi$ is linear or $Z$ is deterministic. By our first assumption we have ruled out the latter. By our second assumption $F_{Q_\theta}(x) \geq F_{\hat{P}_m}(x)$, we can apply Jensen's inequality at every $x$ to deduce that

$$
\begin{aligned}
\mathbb{E}_{\mathbf{X}_m} \left[ \nabla_\theta l_p^p(\hat{P}_m, Q_\theta) \right] &< \nabla_\theta l_p^p(P, Q_\theta), \quad \text{if } 1 < p < 2, \\
\mathbb{E}_{\mathbf{X}_m} \left[ \nabla_\theta l_p^p(\hat{P}_m, Q_\theta) \right] &> \nabla_\theta l_p^p(P, Q_\theta), \quad \text{if } p > 2, \\
\mathbb{E}_{\mathbf{X}_m} \left[ \nabla_\theta l_p^p(\hat{P}_m, Q_\theta) \right] &= \nabla_\theta l_p^p(P, Q_\theta), \quad \text{if } p = 2.
\end{aligned}
$$

We conclude that of the $l_p^p$ distances, only the Cramér distance has unbiased sample gradients. $\qquad\square$

**Proposition 3.** *The energy distance $\mathcal{E}(P, Q)$ has properties (I), (S), and (U).*

*Proof.* As before, write $\mathcal{E}(X, Y) := \mathcal{E}(P, Q)$. Recall that

$$
\mathcal{E}(X, Y) = 2\,\mathbb{E}\,\|X - Y\|_2 - \mathbb{E}\,\|X - X'\|_2 - \mathbb{E}\,\|Y - Y'\|_2 \,.
$$

Consider a random variable $A$ independent of $X$ and $Y$. First, we want to prove property (I):

$$
\mathcal{E}(A + X, A + Y) \leq \mathcal{E}(X, Y).
$$

We will use Proposition 2 from Székely & Rizzo (2013) to express the energy distance in terms of characteristic functions $\phi_X, \phi_Y$ of $d$-dimensional random variables $X$ and $Y$:

$$
\mathcal{E}(X, Y) = \frac{1}{c_d} \int_{R^d} \frac{|\phi_X(t) - \phi_Y(t)|^2}{|t|^{d+1}} dt
$$

where

$$
c_d = \frac{\pi^{(d+1)/2}}{\Gamma(\frac{d+1}{2})}.
$$

The proof then uses properties of characteristic functions ($|\phi_A(t)| \leq 1$ and $\phi_{A+X}(t) = \phi_A(t)\phi_X(t)$ for independent variables $A$ and $X$) to show:

$$
\begin{aligned}
\mathcal{E}(A+X, A+Y) &= \frac{1}{c_d} \int_{R^d} \frac{|\phi_{A+X}(t) - \phi_{A+Y}(t)|^2}{|t|^{d+1}} dt \\
&= \frac{1}{c_d} \int_{R^d} \frac{|\phi_A(t)\phi_X(t) - \phi_A(t)\phi_Y(t)|^2}{|t|^{d+1}} dt \\
&= \frac{1}{c_d} \int_{R^d} \frac{|\phi_X(t) - \phi_Y(t)|^2}{|t|^{d+1}} |\phi_A(t)|^2 dt \\
&\leq \frac{1}{c_d} \int_{R^d} \frac{|\phi_X(t) - \phi_Y(t)|^2}{|t|^{d+1}} dt \\
&= \mathcal{E}(X, Y).
\end{aligned}
$$

This proves (I). Next, consider a real value $c > 0$. We have

$$
\begin{aligned}
\mathcal{E}(cX, cY) &= 2\,\mathbb{E}\,\|cX - cY\|_2 - \mathbb{E}\,\|cX - cX'\|_2 - \mathbb{E}\,\|cY - cY'\|_2 \\
&= 2c\,\mathbb{E}\,\|X - Y\|_2 - c\,\mathbb{E}\,\|X - X'\|_2 - c\,\mathbb{E}\,\|Y - Y'\|_2 \\
&= c\mathcal{E}(X, Y).
\end{aligned}
$$

This proves (S). Finally, suppose that $Y$ is distributed according to $Q_\theta$ parametrized by $\theta$. Let $\mathbf{X}_m = X_1, \ldots, X_m$ be drawn from $P$, and let $\hat{P}_m := \frac{1}{m} \sum_{i=1}^m \delta_{X_i}$. Let $\hat{X}$ be the random variable distributed according to $\hat{P}_m$, and $\hat{X}'$ an independent copy of $\hat{X}$. Then

$$
\mathcal{E}(\hat{P}_m, Q_\theta) = \mathcal{E}(\hat{X}, Y) = 2\,\mathbb{E}\,\big\|\hat{X} - Y\big\|_2 - \mathbb{E}\,\big\|\hat{X} - \hat{X}'\big\|_2 - \mathbb{E}\,\big\|Y - Y'\big\|_2.
$$

The gradient of the true loss w.r.t. $\theta$ is

$$
\nabla_\theta \mathcal{E}(X, Y) = 2\nabla_\theta\,\mathbb{E}\,\big\|X - Y\big\|_2 - \nabla_\theta\,\mathbb{E}\,\big\|Y - Y'\big\|_2. \tag{8}
$$

Now, taking the gradient of the sample loss w.r.t. $\theta$,

$$
\nabla_\theta \mathcal{E}(\hat{X}, Y) = 2\nabla_\theta\,\mathbb{E}\,\big\|\hat{X} - Y\big\|_2 - \nabla_\theta\,\mathbb{E}\,\big\|Y - Y'\big\|_2. \tag{9}
$$

Since the second terms of the gradients match, all we need to show is that the first terms are equal, in expectation. Assuming that $\nabla_\theta$ and the expectation over $\mathbf{X}_m$ commute, we write

$$
\begin{aligned}
\mathop{\mathbb{E}}_{\mathbf{X}_m} \nabla_\theta\,\mathbb{E}\,\big\|\hat{X} - Y\big\|_2 &= \nabla_\theta \mathop{\mathbb{E}}_{\mathbf{X}_m}\,\mathbb{E}\,\big\|\hat{X} - Y\big\|_2 \\
&= \nabla_\theta \mathop{\mathbb{E}}_{\mathbf{X}_m}\,\mathbb{E} \mathop{\mathbb{E}}_{x \sim \hat{P}_m}\,\big\|x - Y\big\|_2,
\end{aligned}
$$

by independence of $X$ and $Y$. But now we know that the expected empirical distribution is $P$, that is

$$
\mathop{\mathbb{E}}_{\mathbf{X}_m} \mathop{\mathbb{E}}_{x \sim \hat{P}_m}\,\mathbb{E}\,\big\|x - Y\big\|_2 = \mathop{\mathbb{E}}_{x \sim P}\,\mathbb{E}\,\big\|x - Y\big\|_2 = \mathbb{E}\,\big\|X - Y\big\|_2.
$$

It follows that the first terms of (8) and (9) are also equal, in expectation w.r.t. $\mathbf{X}_m$. Hence we conclude that the energy distance has property (U), that is

$$
\mathop{\mathbb{E}}_{\mathbf{X}_m \sim P} \nabla_\theta \mathcal{E}(\hat{P}_m, Q_\theta) = \nabla_\theta \mathcal{E}(P, Q_\theta).
$$

$\square$

## B    COMPARISON WITH THE WASSERSTEIN DISTANCE

Figure 2 (left) provides learning curves for the toy experiment described in Section 4.2.

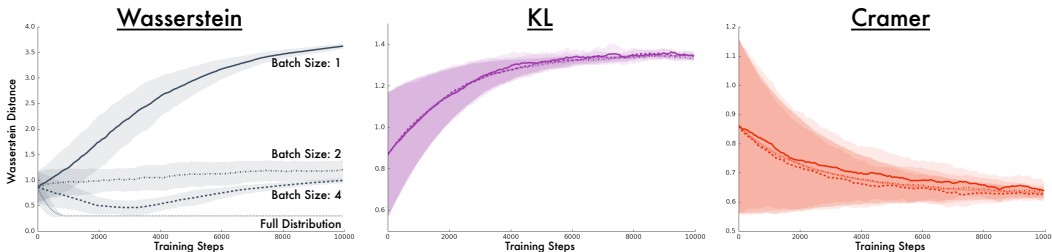

Figure 6: Wasserstein while training to minimize different loss functions (Wasserstein, KL, Cramér). Averaged over 10 random initializations. Error-bands indicate one standard deviation. Note the different y-axes.

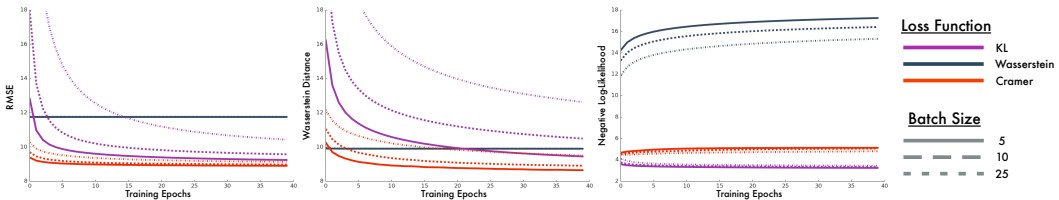

Figure 7: Ordinal regression on the year prediction MSD dataset. Each loss function trained with various minibatch sizes. Training progress shown in terms of: **Left.** RMSE, **Middle.** Wasserstein distance, **Right.** Negative log-likelihood.

## B.1 ORDINAL REGRESSION

We compare the different losses on an ordinal regression task using the Year Prediction MSD dataset from (Lichman, 2013). The task is to predict the year of a song (taking on values from 1922 to 2011), from 90-dimensional feature representation of the song.[4] Previous work has used this dataset for benchmarking regression performance (Hernández-Lobato & Adams, 2015), treating the target as a continuous value. Following Hernández-Lobato & Adams (2015), we train a network with a single hidden layer with 100 units and ReLU non-linearity, using SGD with 40 passes through the training data, using the standard train-test split for this dataset (Lichman, 2013). Unlike (Hernández-Lobato & Adams, 2015), the network outputs a probability distribution over the years (90 possible years from 1922-2011).

We train models using either the 1-Wasserstein loss, the Cramér loss, or the KL loss, the latter of which reduces the ordinal regression problem to a classification problem. In all cases, we compare performance for three different minibatch sizes, i.e. the number of input-target pairs per gradient step. Note that the minibatch size only affects the gradient estimation, but has otherwise no direct relation to the number of samples $m$ previously discussed, since each sample corresponds to a different input vector. We report results as a function of number of passes over the training data so that our results are comparable with previous work, but note that smaller batch sizes get more updates.

The results are shown in Figure 2. Training using the Cramér loss results in the lowest root mean squared error (RMSE) and the final RMSE value of 8.89 is comparable to regression (Hernández-Lobato & Adams, 2015) which directly optimizes for MSE. We further observe that minimizing the Wasserstein loss trains relatively slowly and leads to significantly higher KL loss. Interestingly, larger minibatch sizes do seem to improve the performance of the Wasserstein-based method somewhat, suggesting that there might be some beneficial bias reduction from combining similar inputs. By contrast, using with the Cramér loss trains significantly faster and is more robust to choice of minibatch size.

---

[4]We refer to `https://archive.ics.uci.edu/ml/datasets/YearPredictionMSD` for the details.

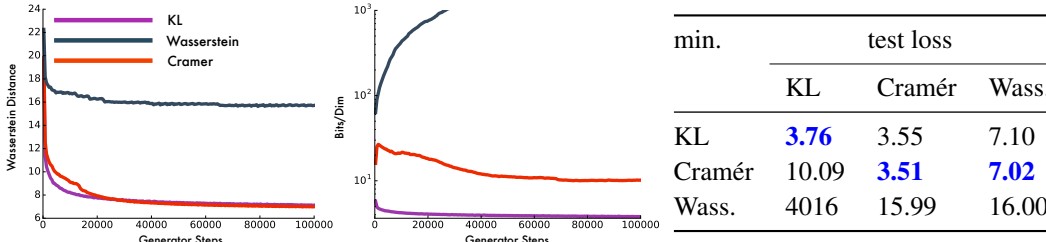

Figure 8: **Left, middle.** Sample Wasserstein and cross-entropy loss curves on the CelebA validation data set. **Right.** Test loss at the end of training, in function of loss minimized (see text for details).

| min. | test loss | | |
|---|---|---|---|
| | KL | Cramér | Wass. |
| KL | **3.76** | 3.55 | 7.10 |
| Cramér | 10.09 | **3.51** | **7.02** |
| Wass. | 4016 | 15.99 | 16.00 |

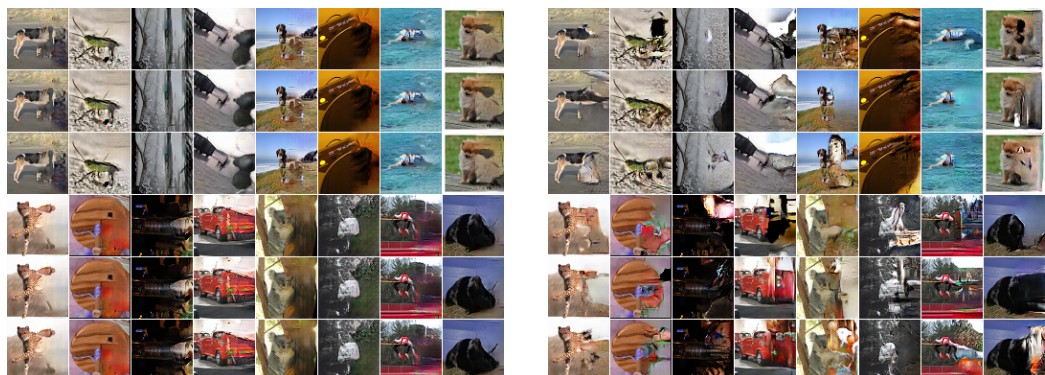

Figure 9: Generated right halves for WGAN-GP (left) and Cramér GAN (right) for left halves from the validation set of Downsampled ImageNet 64x64 (Van den Oord et al., 2016). The low diversity in WGAN-GP samples is consistent with the observations of Isola et al. (2016): *"the generator simply learned to ignore the noise."*

## B.2    IMAGE MODELLING WITH PIXELCNN

As additional supporting material, we provide here the results of experiments on learning a probabilistic generative model on images using either the 1-Wasserstein, Cramér, or KL loss. We trained a PixelCNN model (Van den Oord et al., 2016) on the CelebA 32x32 dataset (Liu et al., 2015), which is constituted of 202,599 images of celebrity faces. At a high level, probabilistic image modelling involves defining a joint probability $Q_\theta$ over the space of images. PixelCNN forms this joint probability autoregressively, by predicting each pixel using a histogram distribution conditional on a probability-respecting subset of its neighbours. This kind of modelling task is a perfect setting to study Wasserstein-type losses, as there is a natural ordering on pixel intensities. This is also a setting in which full distributions are almost never available, because each prediction is conditioned on very different context; and hence we require a loss that can be optimized from single samples. Here the true losses are not available. Instead we report the sample Wasserstein loss, which is an upper bounds on the true loss Bellemare et al. (proof is provided by 2017). For the KL divergence we report the cross-entropy loss, as is typically done; the KL divergence itself corresponds to the expected cross-entropy loss minus the real distribution's (unknown) entropy.

Figure 8 shows, as in the toy example, that minimizing the Wasserstein distance by means of stochastic gradient fails. The Cramér distance, on the other hand, is as easily minimized as the KL and in fact achieves lower Wasserstein and Cramér loss. We note that the resulting KL loss is higher than when directly minimizing the KL, reflecting the very real trade-off of using one loss over another. We conclude that in the context of learning an autoregressive image model, the Cramér should be preferred to the Wasserstein metric.

## C    CRAMÉR GAN

### C.1    LOSS FUNCTION DETAILS

Our critic has a special form:

$$f(x) = \mathop{\mathbb{E}}_{Y' \sim Q} \|h(x) - h(Y')\|_2 - \mathop{\mathbb{E}}_{X' \sim P} \|h(x) - h(X')\|_2$$

where $Q$ is the generator and $P$ is the target distribution. The critic has trainable parameters only inside the deep network used for the transformation $h$. From (4), we define the generator loss to be

$$L_g(X, Y) = \mathop{\mathbb{E}}_{X \sim P}[f(X)] - \mathop{\mathbb{E}}_{Y \sim Q}[f(Y)], \tag{10}$$

as in Wasserstein GAN, except that no $\max_f$ operator is present and we can obtain unbiased sample gradients. At the same time, to provide helpful gradients for the generator, we train the transformation $h$ to maximize the generator loss. Concretely, the critic seeks to maximize the generator loss while minimizing a gradient penalty:

$$L_{critic}(X, Y) = -L_g(X, Y) + \lambda \mathrm{GP} \tag{11}$$

where GP is the gradient penalty from the original WGAN-GP algorithm (Gulrajani et al., 2017) (the penalty is given in Algorithm 1). The gradient penalty bounds the critic's outputs without using a saturating function. We chose $\lambda = 10$ from a short parameter sweep. Our training is otherwise similar to the improved training of Wasserstein GAN (Gulrajani et al., 2017).

In the next two sections, we describe how to practically compute gradients of these losses with respect to the generator and transformation parameters, respectively.

### C.2    GRADIENT ESTIMATES FOR THE GENERATOR

Recall that the energy distance is:

$$\mathcal{E}(X, Y) = 2 \mathop{\mathbb{E}}_{\substack{X \sim P \\ Y \sim Q}} \|X - Y\|_2 - \mathop{\mathbb{E}}_{\substack{X \sim P \\ X' \sim P}} \|X - X'\|_2 - \mathop{\mathbb{E}}_{\substack{Y \sim Q \\ Y' \sim Q}} \|Y - Y'\|_2$$

If $Y$ is generated from the standard normal noise $Z \sim N(0, 1)$ by a differentiable generator $Y = G(Z)$ and the generator has an integrable gradient, we can use the reparametrization trick (Kingma & Welling, 2014) to compute the gradient with respect to the generator parameters:

$$\nabla_{\theta_G} \mathcal{E}(X, Y) = 2 \mathop{\mathbb{E}}_{\substack{X \sim P \\ Z \sim N(0,1)}} \nabla_{\theta_G} \|X - G(Z)\|_2 - \mathop{\mathbb{E}}_{\substack{Z \sim N(0,1) \\ Z' \sim N(0,1)}} \nabla_{\theta_G} \|G(Z) - G(Z)'\|_2.$$

We see that we only need one real sample $X$ to estimate the gradient, because the $\|X - X'\|$ term does not depend on the generator parameters. This allows us to define a generator loss usable for situations with only one real sample (e.g., for conditional modeling):

$$\hat{L}_g(X, Y) = 2 \mathop{\mathbb{E}}_{\substack{X \sim P \\ Y \sim Q}} \|h(X) - h(Y)\|_2 - \mathop{\mathbb{E}}_{\substack{Y \sim Q \\ Y' \sim Q}} \|h(Y) - h(Y')\|_2 \tag{12}$$

### C.3    GRADIENT ESTIMATES FOR THE TRANSFORMATION

As shown in the previous section, we can obtain an unbiased gradient estimate of the generator loss (12) from three samples: two from the generator, and one from the target distribution. However, to estimate the gradient of the Cramér GAN loss with respect to the transformation parameters we need *four* independent samples: two from the generator and two from the target distribution. In many circumstances, for example when learning conditional densities, we do not have access to two independent target samples. We will instead define a surrogate objective for the critic. The surrogate critic will have the following form:

$$f_s(x) = \mathop{\mathbb{E}}_{Y' \sim Q} \|h(x) - h(Y')\|_2 - \|h(x)\|_2$$

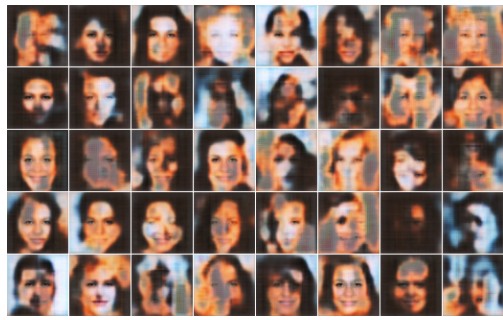 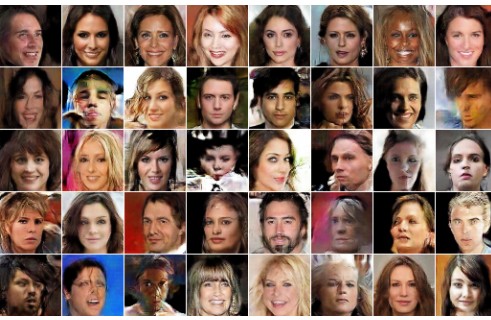

Figure 10: **Left.** Generated images from a generator trained to minimize the energy distance of raw images, $\mathcal{E}(X, Y)$. **Right.** Generated images if minimizing the Cramér GAN loss, $\mathcal{E}(h(X), h(Y))$. Both generators had the same DCGAN architecture (Radford et al., 2015).

which we use to define a surrogate loss $L_s(X, Y)$ similar to (10):

$$
\begin{aligned}
L_s(X, Y) &= \mathop{\mathbb{E}}_{X \sim P}[f_s(X)] - \mathop{\mathbb{E}}_{Y \sim Q}[f_s(Y)] \\
&= \mathop{\mathbb{E}}_{\substack{X \sim P \\ Y' \sim Q}} \|h(X) - h(Y')\|_2 - \mathop{\mathbb{E}}_{X \sim P} \|h(X)\|_2 \\
&\quad - \mathop{\mathbb{E}}_{\substack{Y \sim Q \\ Y' \sim Q}} \|h(Y) - h(Y')\|_2 + \mathop{\mathbb{E}}_{Y \sim Q} \|h(Y)\|_2
\end{aligned} \tag{13}
$$

The surrogate loss emulates an integral probability metric (IPM) (Müller, 1997) and can be used to train the critic. The maximization of this loss will force $\mathbb{E}\|h(X) - h(Y')\|_2$ and $\mathbb{E}\|h(Y) - h(Y')\|_2$ to be informative about the underlying distributions.

The generator can be then trained to minimize the energy distance $\hat{L}_g$ (12) of the transformed variables. It is also possible to obtain training more similar to Wasserstein GAN by training the generator to minimize the surrogate loss (13). We recommend trying both possibilities, because they were both stable and produced diverse conditional samples. The whole training procedure is summarized as Algorithm 1.

Finally, when estimating the losses in Algorithm 1, we use two independent samples $x_g, x_g'$ from the generator. However, in constructing the surrogate loss $\tilde{L}_s$, an asymmetry arises. We reduce variance by averaging the two losses $\tilde{L}_s(x_g, x_g')$ and $\tilde{L}_s(x_g', x_g)$.

### C.4 GENERATOR ARCHITECTURE

The generator architecture is the U-Net (Ronneberger et al., 2015) previously used for Image-to-Image translation (Isola et al., 2016). We used no batch normalization and no dropout in the generator and in the critic. The network conditioned on the left half of the image and on extra 12 channels with Gaussian noise. We generated two independent samples for a given image to compute the Cramér GAN loss. To be computationally fair to WGAN-GP, we trained WGAN-GP with twice the minibatch size (i.e., the Cramér GAN minibatch size was 64, while the WGAN-GP minibatch size was 128).

### C.5 CRITIC ARCHITECTURE

Our $h(x)$ transformation is a deep network with 256 outputs (more is better). The network has the traditional deep convolutional architecture (Radford et al., 2015). We do not use batch normalization, as it would conflict with the gradient penalty.

### C.6 PERFORMANCE EVALUATION

We report the Inception score (Salimans et al., 2016) and the Fréchet Inception Distance (FID) (Heusel et al., 2017) in Figure 11 (left), which are commonly used measures of evaluation for GANs.

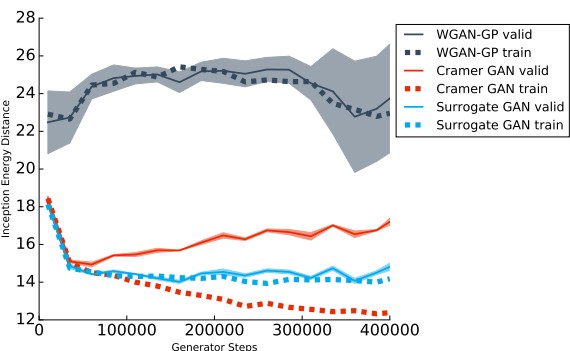

| Model | Inception | FID |
|---|---|---|
| Training set | 11.2 | 0.0 |
| WGAN-GP | 6.5 | 36.4 |
| Cramér GAN | 6.7 | 33.6 |
| Surrogate GAN | 6.6 | 34.1 |

Figure 11: **Left.** Inception score and FID on CIFAR-10. The Surrogate GAN is a Cramér GAN with the generator trained to minimize the surrogate loss (13). **Right.** Inception Energy Distance on conditional CIFAR-10. The network conditioned on the left half of the CIFAR-10 images. The shaded area denotes the standard deviation from 3 runs.

These evaluation measures have the disadvantage that they are not able to detecting overfitting and account for diversity in generated conditional samples. For example, a mixture model that overfits to the training set would get a better Inception score and FID than the trained GANs.

We propose a new evaluation for conditional GANs that uses data from the validation set and that is able to detect overfitting. Our *Inception Energy Distance* (IED) measures a difference, similar to the genererator loss (12), between features of completed image and features of the corresponding real image. An unbiased estimator of the IED is:

$$\text{IED} = \left\| in(x_r) - in(x_g) \right\|_2 + \left\| in(x_r) - in(x'_g) \right\|_2 - \left\| in(x_g) - in(x'_g) \right\|_2 \qquad (14)$$

where $x_r$ is a real sample and $x_g, x'_g$ are two independent generated samples. $in(x)$ are the features for image $x$, and is the is the output of the pretrained Inception network[5] (Szegedy et al., 2016), specifically the output layer `pool_3:0` with 2048 features. The pretrained Inception network allows to objectively compare different GANs. Our performance measure is similar to the FID, but can be computed with one real sample and monitored online.

We use the Inception Energy Distance only to detect underfitting and overfitting. Figure 11 (right) shows that WGAN-GP is not minimizing IED on the training set. WGAN-GP produces very deterministic completions and this is detected by the $\left\| in(x_g) - in(x'_g) \right\|_2$ term in the IED. We also see that the Cramér GAN is overfitting the training set. The Cramér GAN is progressively learning the distribution of the training set and obtains a worse IED on the validation set. This suggests that our optimization is able to successfully train the generator, and that with more data and regularization methods, we will be able to overcome this overfitting. For example, future work can train on large video datasets and try to minimize the IED directly.

---

[5]http://download.tensorflow.org/models/image/imagenet/inception-2015-12-05.tgz