# OpenReview forum: "The Cramer Distance as a Solution to Biased Wasserstein Gradients"
_ICLR.cc/2018/Conference — Reject_

### Official Review · AnonReviewer1 · 2017-11-25
**Nicely written paper, nice review of some interesting properties of divergence measures, narrow scope w.r.t the problem addressed,  and on-the-threshold experiments**

**Rating:** 5
**Confidence:** 3

**Review:**

The manuscript proposes to use the Cramer distance as a measure between distributions (acting as a loss) when optimizing
an objective function using stochastic gradient descent (SGD). Cramer distance is a Bregman divergence and is a member of the Lp family of divergences.  Here a "distance" means a symmetric divergence measure that satisfies the relaxed triangle inequality. The motivation for using the Cramer distance is that it has unbiased sample gradients while still enjoying some other properties such as scale sensitivity and sum invariant. The authors also proof that for the Bernoulli distribution, there is a lower bound independent of the sample size for the deviation between the gradient of the Cramer distance, and the expectation of the estimated gradient of the Cramer distance. Then, the multivariate case of the Cramer distance, called the energy distance, is also briefly presented. The paper closes with some experiments on ordinal regression using neural networks and training GANs using the Cramer distance.

In general, the manuscript is well written and the ideas are smoothly presented. While the manuscript gives some interesting insights, I find that the contribution could have been explained in a more broader sense, with a stronger compelling message.

Some remarks and questions:

1.	The KL divergence considered here is sum invariant but not scale sensitive, and has unbiased sample gradients. The
	authors are considering here the standard (asymmetric) KL divergence (sec. 2.1). Is it the case that losing scale
	sensitivity make the KL divergence insensitive to the geometry of the outcomes? or is it due to the fact the KL
	divergence is not symmetric? or ?

2.	The main argument for the paper is that the simple sample-based estimate for the gradient using the Wasserstein
	metric is a biased estimate for the true gradient of the Wasserstein distance, and hence it is not favored with
	SGD-type algorithms. Are there any other estimators in the literature for the gradient of the Wasserstein distance?
	Was this issue overlooked in the literature?

3.	I am not sure if a biased estimate for the gradient will lead to a ``wrong minimum'' in an energy space that has
	infinitely many local minima.  Of course one should use an unbiased estimate for the gradient whenever this is possible.
	However, even when this is possible, there is no guarantee that this will consistently lead to deeper and ``better''
	minima, and there is no guarantee as well that these deep local minima reflect meaningful results.

4.	To what extent can one generalize theorem 1 to other probability distributions (continuous and discrete) and to the
	multivariate cases as well?

5.	I also don't think that the example given in sec. 4.2 and depicted in Fig. 1 is the best and simplest way to illustrate
	the benefit of Cramer distance over Wasserstein. Similarly, the experiments for the multivariate case using GANs and
	Neural Networks do not really deliver tangible, concrete and conclusive results. Partly, these results are very
       qualitative, which can be understood within the context of GANs. However, the authors could have used other
       models/algorithms where they can obtain concrete quantitative results (for this type of contribution). In addition,
	such sophisticated models (with various hyper-parameters) can mask the true benefit for the Cramer distance, and can
	also mask the extent of how good/poor is the sample estimate for the Wasserstein gradient.

---

> ### Author Response · Authors · 2017-12-11
> **Re: Nicely written paper**
>
> 1. The Cramer and Wasserstein are scale sensitive because they incorporate the Euclidean metric between outcomes. The KL divergence, on the other hand, only compares the relative probability densities.
>
> 2. Quantile regression is one method for minimizing Wasserstein distances which we learned of after completing this work. However and to the best of our knowledge, it is not always applicable, for example in a GAN setting.
>
> 3. We believe the deterministic completions discovered in Wasserstein GAN reflect the issues raised by Theorem 1, i.e. that the wrong minimum is found (in the sense that the true underlying distribution is not deterministic).
>
> 4. We expect an analogue of Theorem 1 to hold whenever we consider asymmetric distributions, both continuous and discrete.
>
> 5. We also trained a PixelCNN model using different loss functions. PixelCNN uses autoregressive univariate distributions to model whole images. There we found that minimizing the sample Wasserstein loss by gradient descent leads to far worse results than when minimizing the Cramer loss, even when the results are measured in terms of the Wasserstein distance itself. The results are summarized in Appendix B.2 and Figure 8. Similar results also held for ordinal regression (Appendix B.1).

---

### Official Review · AnonReviewer2 · 2017-11-30
**The paper proposes a too vague discussion of the (dis-)advantages of metrics in statistical learning, which discussion has already taken place in the 50's-60's in the domain of Statistics**

**Rating:** 4
**Confidence:** 5

**Review:**

The contribution of the article is related to performance criteria, and in particular to the Wasserstein/Mallows metric, which has received a good deal of attention these last few years in the machine learning literature. The paper starts with a discussion about desirable properties of a loss function and points out the fact that (plug-in) empirical versions of the gradient of this quantity are biased, which limits its interest, insofar as many learning techniques are based on (stochastic) gradient descent. In its current state, this argument looks artificial. Indeed, zero bias can be a desirable properties for an estimate but being biased does not prevent it from being accurate. In contrast, in many situations like ridge regression, incorporating bias permits to drastically reduce variance.It quite depends on the structural assumptions made. For this reason, the worst case result (Theorem 1) is not that informative in my opinion. As they are mainly concerned by the L_1 version of the Wasserstein distance, rather than focussing on the bias, the authors could consider the formulation in terms of inverse cumulative distribution functions in the 1-d setup and the fact that the empirical cdf is never invertible: even if the theoretical cdf is invertible (which naturally guarantees uniqueness of the optimal transport) the underlying mass transportation problem is not as well-posed as that related to its statistical counterpart (however, smoothing the empirical distribution may remedy this issue).
The authors propose to use instead the Cramer distance, which is a very popular distance in Statistics and on which many statistical hypothesis testing procedures rely, and review its appealing properties. The comparisons between KL, Wasserstein and Cramer distances is vain in my opinion and willing to come to a general conclusion about the merits of one against the others is naive. In a nonparametric setup, it is always possible to find distributions such that certain of its properties are hidden by certain distances and highlighted by others. This is precisely why you are forced to specify the type of deviations between distributions in nonparametric hypothesis testing (a shift, a change in scale, etc.), there is no way of assessing universally that two distributions are close: optimality can only be assessed for sequences of contiguous hypotheses. The choice of the distance is part of the learning problem.

---

> ### Author Response · Authors · 2017-12-11
> **Re: The paper proposes a too vague discussion**
>
> We thank the reviewer for their points, which are well-taken and will certainly help us improve the paper presentation. In particular, your feedback suggests it would be helpful to emphasize the empirical results currently in the appendix in order to better frame the theory.
>
> There seems to be a misunderstanding regarding Theorem 1: our result shows that bias occurs not just in the gradients but also in the minimizer of the sample Wasserstein loss. Put another way: SGD on this loss, as an estimation procedure, is not consistent. This is fundamentally different from what occurs in ridge regression.
>
> In the appendix we included empirical results on two domains (ordinal regression and image modelling) where we show that minimizing the sample Wasserstein loss by gradient descent leads to far worse results than when minimizing the Cramer loss, even when the results are measured in terms of the Wasserstein distance itself. While it's true that in general no distributional metric dominates the others, here we are highlighting a problem that has visible consequences.
>
> Regarding the merits of comparing to the KL divergence: we acknowledge the point, but do not completely agree. The shift in machine learning in the last years has exactly been recognizing that the KL divergence is ill-suited in many problems of current interest (generative modelling, cost-sensitive classification, reinforcement learning…). Our aim here was to illustrate the qualitative similarities between Cramer and Wasserstein distances, compared to the KL divergence.

---

### Official Review · AnonReviewer3 · 2017-12-01
**Nice read; provides some understanding for GAN training**

**Rating:** 7
**Confidence:** 2

**Review:**

The authors investigate how the properties of different discrepancies for distributions affect the training of parametric model with SGD. They argue that in order for SGD to be a useful training procedure, an ideal metric should be scale sensitive, sum invariant and also provide unbiased gradients. The KL-divergence is not is scale sensitive, and the Wasserstein metric does not provide unbiased gradients. The authors thus posit the Cramer distance as a foundation for the discriminator in the GAN, and then generalize this to an energy based discriminator. The authors then test their Cramer GAN on the CelebA dataset and show comparable results to the Wasserstein GAN, with less mode collapse.

From what I can gather, the Cramer GAN is unlikely to be a huge improvement in the GAN literature, but the mathematical relationships investigated in the paper are illuminating. This brings some valuable understanding for improving upon previous GANs [e.g. WGAN]. As energy based GANs and MMD GANs have become more prominent, it would be nice to see how these ideas interplay with those GANs. However, overall I thought the paper did a nice job presenting some context for GAN training.

---

> ### Author Response · Authors · 2017-12-11
> **Re: Nice read; provides some understanding for GAN training**
>
> Thank you for your comments; we agree there is a need for unifying some of the GAN literature into a coherent story.

---

### Public Comment · (anonymous) · 2017-11-05
**Some subtleties of comparing Wasserstein distance and energy distance**

There are some subtleties I find the authors may need to clarify a bit more:

In the formulation of W-GAN, the critic/discriminator is essentially part of the definition of Wasserstein distance -- a parametric approximation for the dual potential. That is, W-GAN is a generative model that "attempts" to minimize the Wasserstein distance between real distribution and generated distribution. W-GAN is not a GAN with Wasserstein loss.

Whereas, in the authors' proposed approach, the concepts of GAN (two-player game) are unavoidable. The use of energy distance creates a surrogate loss function that discriminator ultimately wants to maximize.

This is part I don't understand why the GAN with energy distance can compare directly to Wasserstein GAN from the model perspective. The way they use distance is by nature different.

My opinion: While I agree that the W-GAN and the approach by authors share many similarities in algorithm, they actually follow different modeling framework to use the distance. The claim that Wasserstein distance does not have unbiased sample gradients is in fact also applicable to the approach by authors if one consider the maximized quantity by critic as a loss for training generator.

---

> ### Author Response · Authors · 2017-12-11
> **Re: Some subtleties**
>
> Thank you for the insightful comment. You’re right that there is a subtle difference between the two approaches. To paraphrase your comment, the WGAN equivalent with Cramer distance would have a critic that learns the function f* itself (Equation 4).
>
> One resulting difference is that if we stop training the WGAN critic, the WGAN generator will  collapse to the single point with the maximal critic value. On the other hand, if we stop training the Cramer GAN critic, the Cramer GAN generator will learn to minimize the energy distance between the distributions of the critic outputs. The Cramer GAN generator will then learn a distribution, not a single point.
>
> You can see in Figure 4 that WGAN is much worse if we do just one critic update per generator update. It is still helpful to train the Cramer GAN critic, to avoid ignoring information not originally present in the critic outputs.

---

### Decision · Program_Chairs · 2018-01-29
**ICLR 2018 Conference Acceptance Decision**

**Decision:**

Reject

**Comment:**

Pros:
- The authors propose a new algorithm to train GAN based on Cramer distance arguing that this eases optimization compared to Wasserstein GAN.
-  Reviewers agree that the paper reads well and provides a good overview of the properties of divergence measures used for GAN training.

Cons:
- It is not clear how much the central arguments about scale sensitivity, sum invariance, and unbiased sample gradients of the distances hold true in practice and generalize.
- The reviewers do not agree the benefits of the new algorithm is clear from the experiments shown.
Given the pros/cons ,the committee feels the paper falls short of acceptance in its current form.